# Understanding the Nonlinear Response of SiPMs

**DOI:** 10.3390/s24082648

**Published:** 2024-04-21

**Authors:** Víctor Moya-Zamanillo, Jaime Rosado

**Affiliations:** IPARCOS-UCM, Instituto de Física de Partículas y del Cosmos, and EMFTEL Department, Universidad Complutense de Madrid, E-28040 Madrid, Spain

**Keywords:** silicon photomultipliers, SiPM, nonlinearity, Monte Carlo simulation, statistical model, correlated noise

## Abstract

A systematic study of the nonlinear response of Silicon Photomultipliers (SiPMs) was conducted through Monte Carlo (MC) simulations. The MC code was validated against experimental data for two different SiPMs. Nonlinearity mainly depends on the balance between the photon rate and the pixel recovery time. Additionally, nonlinearity has been found to depend on the light pulse shape, the correlated noise, the overvoltage dependence of the photon detection efficiency, and the impedance of the readout circuit. Correlated noise has been shown to have a minor impact on nonlinearity, but it can significantly affect the shape of the SiPM output current. Considering these dependencies and a previous statistical analysis of the nonlinear response of SiPMs, two phenomenological fitting models were proposed for exponential-like and finite light pulses, explaining the roles of their various terms and parameters. These models provide an accurate description of the nonlinear responses of SiPMs at the level of a few percentages for a wide range of situations.

## 1. Introduction

Silicon photomultipliers (SiPMs) are solid-state photodetectors that offer excellent characteristics, including high gain, fast timing properties, very good photon-counting resolution, and high quantum efficiency [1]. Additionally, they are compact, relatively affordable, operate at low voltage (a few tens of volts), and insensitive to magnetic fields. Due to their numerous advantages over classical photomultiplier tubes (PMTs), SiPMs are increasingly utilized in various fields, such as high-energy physics experiments [2,3], medical imaging and dosimetry [4,5], biophotonics [6], light detection and ranging (LiDAR) systems [7], and more.

A SiPM consists of an array of Geiger-mode avalanche photodiodes (G-APDs), hereafter called pixels, connected in parallel to a common readout (see Figure 1) so that the SiPM output signal is the sum of the signals from all the pixels. The device is biased above the breakdown voltage, with the difference between the bias voltage and the breakdown voltage referred to as overvoltage. This configuration enables a single incident photon to trigger a high-multiplication breakdown avalanche, resulting in a measurable signal. In ideal conditions, when a light pulse illuminates a SiPM, the output charge should be directly proportional to the number of detected photons. However, SiPMs exhibit a nonlinear response when the number of impinging photons is comparable to the number of pixels of the device. Significant efforts are being carried out to extend the linear dynamic range of SiPMs (see, e.g., [8,9]).

Nonlinearity in a SiPM depends on the time it takes for the overvoltage of a pixel to recover after a breakdown avalanche. A fraction of photons may interact with unrecovered pixels, which have a lower trigger probability and gain, resulting in a SiPM signal with a lower amplitude than expected. The pixel overvoltage can also be reduced due to the voltage drop across the readout-circuit resistance [10]. Therefore, nonlinearity depends on the photon rate (i.e., both the amplitude and width of the light pulse) and the recovery time of pixels in a complex way. Additionally, SiPMs exhibit both correlated and uncorrelated noise, further complicating their performance [11,12,13].

Due to the complexity of the problem, an exact analytical treatment of the nonlinear response of SiPMs is infeasible. Nevertheless, there exist several statistical models that rely on various simplifications of the SiPM response [14,15,16,17,18]. In [14], a comprehensive statistical model of the SiPM response to light pulses of arbitrary shape and duration was presented. This model provides a simple expression for the mean output charge of a SiPM, which depends on only two parameters that can be measured or fitted. However, due to the adopted approximations, the applicability of this model is limited to situations where nonlinear effects and correlated noise are moderate. Furthermore, the model cannot describe the statistical fluctuations of the output charge or nonlinear effects on the signal shape.

A Monte Carlo (MC) treatment can potentially include all factors that affect the SiPM performance, allowing a detailed simulation of the response of any SiPM by utilizing appropriate input parameters. Various MC codes are available on different platforms [19,20,21,22,23,24]. The Matlab MC code developed by Abhinav et al. [24] is especially complete, as it simulates the triggering of breakdown avalanches from photons and noise on an individual pixel basis, including the pixel recovering and the reduction in the pixel overvoltage due to the voltage drop across the readout-circuit resistance. Therefore, this code is very suitable for simulating nonlinear effects. Nevertheless, several relevant aspects of the SiPM performance are treated approximately, particularly the description of the correlated noise and the trigger probability of recovering pixels. On the other hand, the code uses a very detailed electrical model to simulate the transient output response due to an avalanche, which requires a relatively large number of input parameters, and some of them are usually not known with sufficient precision.

In this work, a hybrid strategy was followed. We used the MC code developed in [24] to conduct a systematic analysis of the different factors affecting the nonlinear responses of SiPMs, regarding both the output charge and the signal shape. To this end, we modified this code to improve the description of the correlated noise and the trigger probability of recovering pixels. In addition, we implemented new light pulse shapes and a simplified electrical model to more easily identify the main parameters on which nonlinearity depends and to understand their role. Secondly, based on a previous statistical model described in [14], we found phenomenological analytical expressions that fit the simulation results of the mean output charge for light pulses of different shapes and arbitrary intensity over a very wide range of SiPM parameters (e.g., overvoltage, correlated noise, and photon detection efficiency). The proposed models provide a simple but accurate description (at the level of a few percentages) of the SiPM response in the nonlinear region, clearly showing the relationships among the many variables of the problem.

The rest of this paper is structured as follows. In Section 2, the SiPM performance and its parameterization are described. In Section 3, the upgraded MC simulation validation with experimental data is discussed, and the different factors affecting the nonlinearity are identified and evaluated. In Section 4, we present our analytical models for nonlinearity, discussing the meaning of their several terms and fitting parameters. The conclusions are presented in Section 5.

## 2. Overview of the SiPM Response

### 2.1. Output Signal

Let us assume a SiPM biased at a certain voltage $V_{\mathrm{bias}}=V_{\mathrm{br}}+U_{\mathrm{op}}$, where $V_{\mathrm{br}}$ is the breakdown voltage, and $U_{\mathrm{op}}$ is the operation overvoltage. When the SiPM is in a steady state, i.e., no signal is produced, all the pixels have the same overvoltage $U_{\mathrm{op}}$. If a photon interacts with a pixel, it can trigger a self-sustaining charge avalanche in the diode PN junction. The charge avalanche rapidly grows, generating a current *I* across the pixel, which comprises a quenching resistor $R_{q}$ placed in series with the diode (see Figure 1). As *I* increases, the voltage across $R_{q}$ increases, while the voltage across the diode decreases. When the diode overvoltage *U* reaches a value close to 0, the avalanche stops. The current continues to flow until *U* is restored to $U_{\mathrm{op}}$. The diode overvoltage as a function of the time elapsed from the photon arrival is therefore given by
(1)$$U\left(t\right)=U_{\mathrm{op}}-I\left(t\right)\;R_{q}\;.$$

The avalanche duration is sub-nanoseconds, while the recovery of the diode overvoltage typically takes a few tens of nanoseconds. When assuming that the avalanche is instantaneous and *U* drops to 0 at that moment, the current flowing through the pixel is
(2)$$I\left(t\right)=\frac{U_{\mathrm{op}}}{R_{q}}\;\exp\left(-\frac{t}{\tau_{\mathrm{rec}}}\right)\;.$$The time constant $\tau_{\mathrm{rec}}$, called the recovery time, is approximately given by $R_{q}C_{d}$, where $C_{d}$ is the diode capacitance. Therefore, the mean total charge of an avalanche is
(3)$$\langle Q_{1}\rangle =C_{d}U_{\mathrm{op}}\;.$$

If $\langle Q_{1}\rangle$ and $R_{q}$ are known, then the recovery time can be obtained as
(4)$$\tau_{\mathrm{rec}}=\frac{R_{q}\langle Q_{1}\rangle }{U_{\mathrm{op}}}\;.$$

The gain of the SiPM is defined as the avalanche multiplication
(5)$$G=\frac{\langle Q_{1}\rangle }{e}\;,$$
where *e* is the elementary charge. Its typical value is around $10^{5}-10^{6}$. The gain is customarily characterized as a function of $U_{\mathrm{op}}$ under single-photon conditions. If a photon triggers an avalanche in a pixel with $U<U_{\mathrm{op}}$, i.e., while it is recovering from a previous avalanche, *U* drops to 0, *I* rises to $U_{\mathrm{op}}/R_{q}$ again, and the pixel recovery restarts. Therefore, the multiplication of the second avalanche is
(6)$$G^{\prime }=\frac{C_{d}U}{e}=G\frac{U}{U_{\mathrm{op}}}\;.$$

The output current of a SiPM is the sum of the currents of all the individual pixels, as illustrated in Figure 2. When the SiPM is illuminated with light pulses, the signal is usually integrated to measure the output charge *Q* for each pulse.

A detailed electrical model of a SiPM, such as the one described in [24], should include the rise and decay times of the charge avalanche, the parasitic capacitance of the quenching resistors, and their connections to the silicon substrate, as well as the input impedance of the readout electronics. As a consequence, the shape of the transients is not a simple exponential function, as assumed by Equation (Equation 2), but it can be modeled using multiple exponential terms [25]. In principle, the output charge *Q* is independent of the shape of the transients. However, in ref. [10,24] the input impedance $R_{s}$ of the readout electronics (e.g., a shunt resistor placed in parallel with the pre-amplifier, as shown in Figure 1) was shown to have an influence on nonlinearity in SiPMs with a large number of pixels ($N_{\mathrm{pix}}≳1000$). Notice that the voltage drop across $R_{s}$ may be significant when $R_{s}$ (typically ∼10Ω) is not negligible compared to the overall resistance due to the quenching resistors ($R_{q}∼100\;k\Omega$) of thousands of pixels in parallel.

### 2.2. Photon Detection Efficiency

The probability that an incident photon causes a charge avalanche is called photodetection efficiency (PDE), which can be approximately described as the product of the fill factor, the quantum efficiency, and the avalanche trigger probability. The fill factor is the ratio of the sensitive area of the SiPM (i.e., excluding the inactive regions between pixels) to its total area. The quantum efficiency is the probability that a photon incident in a pixel creates an electron-hole pair inside the depleted volume around the PN junction, which is a function of wavelength. The avalanche trigger probability is the probability that free carriers generated in the depleted volume cause a successful charge avalanche. This probability depends on $U_{\mathrm{op}}$ and the depth at which the photon is absorbed; hence, it depends on wavelength.

In this work, we assumed the following expression for the PDE at a given wavelength:(7)$$\varepsilon \left(U_{\mathrm{op}}\right)=\left\{\begin{array}{cc} \varepsilon_{\max}\;\left[1-\exp\left(-\frac{U_{\mathrm{op}}-U_{0}}{U_{\mathrm{ch}}}\right)\right] & \mathrm{if}\;U_{\mathrm{op}}>U_{0}\\ 0 & \mathrm{if}\;U_{\mathrm{op}}\leq U_{0}\;, \end{array}\right.$$
which generally fits the experimental data of typical blue-sensitive SiPMs well (see, e.g., [26]). In this equation, $\varepsilon_{\max}$ is the saturation PDE value, $U_{\mathrm{ch}}$ is a characteristic overvoltage value that determines how fast the PDE grows with $U_{\mathrm{op}}$, and $U_{0}$ is a small overvoltage threshold that has to be included to fit the PDE data for some SiPMs (e.g., $U_{0}=0.66$ V at 405 nm for the Hamamatsu S13360-1325CS SiPM [14]). More sophisticated models of the PDE proposed in [27,28] may be used instead of Equation (Equation 7) to fit experimental data if needed, especially for red- and NIR-sensitive SiPMs.

As with the gain, the PDE is characterized at low-light intensity conditions, at which all the pixels can be assumed to be in a steady state. However, the trigger probability in a recovering pixel with overvoltage $U<U_{\mathrm{op}}$ is smaller. We assumed that the trigger probability varies with *U* in the same fashion as the PDE varies with $U_{\mathrm{op}}$, as given by Equation (Equation 7). Notice that the trigger probability is not proportional to *U*, unlike the avalanche multiplication in Equation (Equation 6).

### 2.3. Parameterization of the Response

It is customary to characterize the SiPM response in terms of $\langle Q\rangle /\langle Q_{1}\rangle$ (i.e., the effective number of full-charge avalanches), as a function of the number of avalanche *seeds*:(8)$$N_{\mathrm{seed}}=\varepsilon \left(U_{\mathrm{op}}\right)N_{\mathrm{ph}}\;,$$
where 〈Q〉 is the mean output charge for light pulses with a given number of photons $N_{\mathrm{ph}}$.

Although an avalanche seed is just a mathematical construct, we can assume that any photon can generate an avalanche seed with a fixed probability $\varepsilon (U_{\mathrm{op}})$, regardless of whether it interacts with a recovered pixel or an unrecovered pixel. Therefore, a seed generated in a pixel with overvoltage *U* has a probability of triggering a charge avalanche:(9)$$P_{\mathrm{trig}}\left(U\right)=\frac{\varepsilon (U)}{\varepsilon (U_{\mathrm{op}})}\;.$$

Ideally, $\langle Q\rangle /\langle Q_{1}\rangle =N_{\mathrm{seed}}$ holds when all the seeds are generated in fully recovered pixels and there is no noise, but $\langle Q\rangle /\langle Q_{1}\rangle <N_{\mathrm{seed}}$ when nonlinearity is significant. In addition, we define two new variables: (10)$$\begin{array}{cc} y & =\frac{\langle Q\rangle }{N_{\mathrm{pix}}\langle Q_{1}\rangle }\;, \end{array}$$(11)$$\begin{array}{cc} x & =\frac{N_{\mathrm{seed}}}{N_{\mathrm{pix}}}\;. \end{array}$$

Notice that both *y* and *x* are independent of $N_{\mathrm{pix}}$ and *G*, allowing for comparisons of the responses of different SiPMs or of the same device at different $U_{\mathrm{op}}$ values.

In the case of light pulses with a duration much shorter than $\tau_{\mathrm{rec}}$ (i.e., no pixel recovery between two consecutive avalanches in a pixel), assuming that photons are uniformly distributed among all the pixels of the SiPM and ignoring noise, the SiPM response is given by the fraction of pixels that are fired, i.e., those with one or more seeds:(12)$$y_{\mathrm{fired}}=1-e^{-x}\;.$$

However, for light pulses with a duration comparable to $\tau_{\mathrm{rec}}$, the relationship between *y* and *x* is much more complex, and *y* can be greater than the unity, because a pixel can be triggered several times during the light pulse.

### 2.4. Noise Effects

SiPMs present both correlated and uncorrelated noise, which produce charge avalanches that are indistinguishable from the ones produced by photons, making the SiPM response even more complex. Uncorrelated noise is caused by thermally generated electron-hole pairs inside the pixel-active region. In our approach, this can be modeled as a Poissonian generation of seeds uniformly distributed among all the pixels. However, the uncorrelated noise rate per pixel is typically below kHz, which is much smaller than $1/\tau_{\mathrm{rec}}$. Therefore, this effect generally has a negligible contribution to the nonlinear responses of SiPMs.

On the other hand, correlated noise is the stochastic production of secondary avalanches induced by a primary avalanche. This effect scales with the signal amplitude and thus might be relevant to nonlinearity. There are three components of correlated noise (see Figure 2). Prompt crosstalk is the nearly simultaneous production of secondary avalanches in neighboring pixels to the pixel where the primary avalanche took place, and it is due to the emission of infrared photons in the primary avalanche (see [12] and references therein). Delayed crosstalk is caused by charge carriers produced by infrared photons in the non-depleted region, which diffuse in the silicon until they reach a neighboring pixel, triggering secondary avalanches with some delay with respect to the primary avalanche [11]. Finally, afterpulsing is similar to delayed crosstalk, but the secondary avalanches are produced in the same pixel where the primary avalanche took place, usually while it is still recovering.

A full-charge avalanche is assumed to produce a random number of seeds of each noise component *i* (ct, dct, or aft) following a Poisson distribution with a mean of $\lambda_{i}$ proportional to $U_{\mathrm{op}}$, because the number of emitted infrared photons is proportional to the avalanche multiplication. In the case that an avalanche is produced in a recovering pixel with overvoltage *U*, the mean number of seeds of each noise component is
(13)$$\lambda_{i}^{\prime }=\lambda_{i}\frac{U}{U_{\mathrm{op}}}\;,$$
which is consistent with Equation (Equation 6). Avalanches due to correlated noise can produce further seeds through correlated noise in a cascade process. Notice that the variables $N_{\mathrm{seed}}$ and *x* are defined without including seeds due to correlated noise. However, correlated noise does contribute to the output charge. Consequently, *y* may be greater than *x* when correlated noise is significant and nonlinearity is low.

For both crosstalk components, the generated seeds are assumed to be uniformly distributed among the four closest neighbors of the primary pixel, which is an approximation that generally reproduces the experimental data well, as shown in [11,12]. For prompt crosstalk, the small delay of the generated seeds with respect to the primary avalanche can be ignored. On the other hand, according to [11], the delay time distribution for afterpulsing and delayed crosstalk is given by
(14)$$\frac{d\lambda_{i}}{dt}\left(t\right)=\left\{\begin{array}{cc} C_{i}\;t^{a_{i}}\;\exp\left(-\frac{t}{\tau_{\mathrm{bulk}}}\right) & \mathrm{if}\;t>t_{\min}\\ 0 & \mathrm{if}\;t\leq t_{\min}\;, \end{array}\right.$$
where $\tau_{\mathrm{bulk}}$ is the minority carrier lifetime in the silicon bulk, which may range from a few ns to hundreds of ns, $a_{\mathrm{aft}}=-1$ for afterpulsing, $a_{\mathrm{dct}}=-0.5$ for delayed crosstalk, $C_{i}$ is a normalization constant such that
(15)$$\lambda_{i}=\int_{t_{\min}}^{\infty }\frac{d\lambda_{i}}{dt}\left(t\right)\;dt\;,$$
and $t_{\min}$ is the minimum time delay between two avalanches that the readout circuit can distinguish, which is typically around 5 ns. This time distribution assumes that afterpulsing is mainly caused by charge carriers diffusing in the silicon bulk, as discussed in [11], but afterpulsing can also be due to the trapping and subsequent release of charge carriers in the depleted volume, resulting in a delay time distribution that can be generally modeled as the sum of exponential functions (see [24] and references therein).

A seed due to crosstalk or afterpulsing is assumed to be able to trigger an avalanche in a recovering pixel with the probability of Equation (Equation 9), just as a seed due to a photon. Notice that if the PDE is assumed to be proportional to $U_{\mathrm{op}}$, the number of correlated avalanches would grow quadratically with $U_{\mathrm{op}}$, as is commonly assumed in the literature. Actually, the overvoltage dependence of the trigger probability of seeds due to correlated noise will differ from that of seeds due to photons, because of their different depth distributions, resulting in distinct $U_{\mathrm{ch}}$ and $U_{0}$ values for each correlated noise component. Nevertheless, this approximation should be sufficient if the correlated noise is not too high.

Correlated noise is usually characterized at single-photon conditions, measuring the probability $P_{i}$ that a primary avalanche produces one or more secondary avalanches of each type. For both crosstalk components, this probability is related to $\lambda_{i}$ by
(16)$$P_{i}=1-e^{-\lambda_{i}}\;.$$

On the other hand, the trigger probability of an afterpulsing seed is a function of the delay time, because the pixel is recovering from the primary avalanche. The afterpulsing probability is related to $\lambda_{\mathrm{aft}}$ by
(17)$$P_{\mathrm{aft}}=1-\exp\left[-\int_{t_{\min}}^{\infty }\frac{d\lambda_{\mathrm{aft}}}{dt}\left(t\right)\;P_{\mathrm{trig}}\left(U\left(t\right)\right)\;dt\right]\;.$$

Therefore, the normalization constant $C_{\mathrm{aft}}$ and hence $\lambda_{\mathrm{aft}}$ can be obtained from the measured $P_{\mathrm{aft}}$ value.

## 3. Simulation Results

As mentioned above, in this work we used the MC code developed in [24], with the following changes:The code was originally designed to simulate scintillation pulses, modeled as double exponential pulses. We also implemented rectangular, triangular, and exponential light pulses to study the effect of the pulse shape on the nonlinear responses of SiPMs.We provided a new option to utilize either the SiPM electrical model developed in [24] or the simplified model described in Section 2.1.In the original code, the probability that a seed triggers an avalanche is modeled as $P_{\mathrm{trig}}\left(U\right)=U/U_{\mathrm{op}}$ (default option) or using PDE data from an external file. We modified the default option so that $P_{\mathrm{trig}}\left(U\right)$ is modeled as in Equation (Equation 9), where the overvoltage dependence of the PDE was assumed to be given by Equation (Equation 7). Therefore, we introduced the new input parameters $U_{\mathrm{ch}}$ and $U_{0}$.In the original code, delayed crosstalk was not included, and both prompt crosstalk and afterpulsing were treated in a simplified way. In particular, the mean number of seeds $\lambda^{\prime }$ produced by an avalanche in a recovery pixel was mistakenly assumed to decrease quadratically with the avalanche multiplication, leading to a slight underestimation of the correlated noise when nonlinearity is significant. We addressed this error so that $\lambda^{\prime }$ is given by Equation (Equation 13). We also improved some computational details in the treatment of correlated noise.The time distribution of afterpulsing was modeled as a sum of exponential functions in the original code. We provided a new option of using Equation (Equation 14) instead.

Our modified version of the code is available upon request.

### 3.1. Experimental Validation

In [24], simulation results were compared with experimental data from [18]. In their setup, several isotopes that emit gamma-ray photons at ten different energies between 27.3 and 1836 keV were used. The gamma-ray photons were incident on two LaBr3:5%Ce scintillators (light yield of 70 ph/keV and decay time of 16 ns) optically coupled to their respective S10362-33-050C SiPMs from Hamamatsu Photonics K.K. [29] (pixel size of 50μm and $N_{\mathrm{pix}}=3600$). The signal from each SiPM was fed into an amplifier with a shunt resistor ($R_{s}=15\;\Omega$), and the resulting output charge was measured. The SiPM parameters needed in the simulations, excluding the specific parameters for the electrical model, which were estimated as described in [24], are summarized in Table 1. The recovery time $\tau_{\mathrm{rec}}$ can be estimated from Equation (Equation 4), resulting in 15.3±1.0 and 12.8±1.0 ns, respectively. Notice that the decay time of the scintillators is similar to $\tau_{\mathrm{rec}}$. To account for the trigger probability $P_{\mathrm{trig}}$ in recovering pixels, Equation (Equation 9), the overvoltage dependence of the PDE was taken from [30]. Measurements of crosstalk did not differentiate between prompt and delayed crosstalk; consequently, the overall crosstalk was treated as prompt crosstalk in the simulations. The delay time distribution of afterpulsing was described using a single exponential function with time constant $\tau_{\mathrm{aft}}$. It should be highlighted that the conversion factor from the number of emitted photons to the number of seeds $N_{\mathrm{seed}}$ (i.e., the product of the photon collection efficiency and the SiPM PDE at the nominal overvoltage $U_{\mathrm{op}}$) was not measured directly, but it was estimated for each detector to match the simulated and experimental data at 27.3 keV. The resulting range of $N_{\mathrm{seed}}$ is from 300 to 21000 (i.e., *x* ranges from 0.1 to 5.8). Both experimental and simulation data extracted from Figure 7 of [24] are reproduced in Figure 3, where we express the results in terms of the variable *y* as a function of the number of scintillation photons. The error bars of the experimental data represent the systematic uncertainty due to the gain measurement. The simulation results are almost identical for both detectors and are illustrated as a single dashed line.

We simulated the experimental case described above using our modified version of the MC code. Our results are depicted as continuous lines in Figure 3. We employed the same input parameters as in [24], including those for the electrical model and the conversion factors fε. The treatment of the correlated noise in recovering pixels differed in our simulations but was found to have a minor effect. Additionally, our simulations vary in the trigger probability of recovering pixels, since we assumed that the overvoltage dependence of the PDE follows Equation (Equation 7), where the parameters $U_{\mathrm{ch}}=1.08$ V and $U_{0}=0.23$ V were obtained from the PDE data at 408 nm provided by the manufacturers. Our simulation results coincide with those from [24] at 27.3 keV, confirming the consistency of the simulations in the linear regime. However, our results are closer to the experimental data at larger energies. This is due to our different treatment of the trigger probability of recovering pixels. In fact, the authors of the original MC code observed that their simulation results matched experimental data well when using a different assumption of the variation in the trigger probability with overvoltage [24]. The color bands represent the variation in our simulation results when both the conversion factor fε and the recovery time are varied consistently within the confidence interval of the gain. Results were obtained by averaging over a large number of simulations to smooth out statistical fluctuations.

In addition, we simulated a similar experiment described in [14]. In this setup, gamma-ray photons with energies from 300 to 2100 keV were incident on a LYSO(Ce) scintillator from Epic Crystal [31] (light yield of 29 ph/keV and decay time of 42 ns) coupled to a Hamamatsu S13360-1350CS SiPM (pixel size of 50μm and $N_{\mathrm{pix}}=667$). This SiPM is distinguished by its low correlated noise (<5% at $U_{\mathrm{op}}=3$ V) and an expanded operational overvoltage range. Measurements were carried out at $U_{\mathrm{op}}$ values from 1 to 11 V. In the simulations, we utilized the simplified electrical model described in Section 2.1 since, for this SiPM with a relatively small number of pixels, the influence of the input impedance of the readout circuit and the parasitic capacitance of the SiPM’s circuit elements were found to be negligible. The recovery time ($\tau_{\mathrm{rec}}=29$ ns) and the prompt crosstalk probability $P_{\mathrm{ct}}$ at $U_{\mathrm{op}}$ values up to 7 V were taken from [11]. No direct measurements of $P_{\mathrm{ct}}$ are available at higher $U_{\mathrm{op}}$ values because correlated noise increases dramatically. Therefore, $P_{\mathrm{ct}}$ values at $U_{\mathrm{op}}=9$ and 11 V were adjusted so that the simulation results fit the experimental data. For simplicity, we neglected delayed crosstalk and assumed $P_{\mathrm{aft}}=P_{\mathrm{ct}}$ for all $U_{\mathrm{op}}$ values. The time distribution of afterpulsing was modeled according to Equation (Equation 14) with $\tau_{\mathrm{bulk}}=8.5$ ns, as reported in [11]. To model $P_{\mathrm{trig}}$ in recovering pixels, we used $U_{\mathrm{ch}}=2.68$ V and $U_{0}=0$ V, as reported in [14]. Similar to the previous case, we determined the conversion factor fε to match the simulated and experimental data for each $U_{\mathrm{op}}$ value. To do this, ε was forced to vary with $U_{\mathrm{op}}$ according to Equation (Equation 7), with the above values of $U_{\mathrm{ch}}$ and $U_{0}$. The resulting conversion factors were consistent with those obtained in [14]. For instance, we obtained fε=0.081 at $U_{\mathrm{op}}=11$ V; hence, the largest gamma-ray energy of 2100 keV corresponds to $N_{\mathrm{seed}}=4900$ (i.e., x=7.4).

A comparison between simulated and experimental data is shown in Figure 4, where the error bars of experimental data (only shown for $U_{\mathrm{op}}=1$ and 11 V) correspond to an estimated systematic uncertainty of 10% due to the gain measurement. The simulation results exhibit the correct nonlinear behavior and dependence on $U_{\mathrm{op}}$. The relatively larger gap between the curves for $U_{\mathrm{op}}=1$ V and $U_{\mathrm{op}}=3$ V is due to a more pronounced increase in the PDE with $U_{\mathrm{op}}$ for $U_{\mathrm{op}}≲U_{\mathrm{ch}}$. The gap between the curves for $U_{\mathrm{op}}=9$ V and $U_{\mathrm{op}}=11$ V is due to a large increase in the correlated noise, where the estimated $P_{\mathrm{ct}}$ values are as high as 24% and 85%, respectively. Discrepancies with respect to the experimental data were smaller than 10% and are likely attributable to errors in the parameters $\tau_{\mathrm{rec}}$ and $U_{\mathrm{ch}}$. As discussed in [14], $U_{\mathrm{ch}}$ was obtained from PDE data at 450 nm, not for the specific emission spectrum of the LYSO(Ce) scintillator.

### 3.2. Photon Rate

Nonlinearity essentially depends on the balance between the rate of photons per pixel and the pixel recovery time $\tau_{\mathrm{rec}}$. If the photon rate is high enough so that photons may arrive at a pixel while it is recovering from a previous avalanche, they will have a reduced probability of triggering an avalanche, and if they do, the avalanche multiplication will be smaller. Even though the number of photons per pixel is large, if the PDE is low or the light pulse is long enough for the average time between consecutive seeds generated in the same pixel to exceed the pixel recovery time, the nonlinearity will be weak. Actually, the relevant parameters are the number of seeds per pixel *x*, Equation (11), and the ratio between the pulse width and $\tau_{\mathrm{rec}}$.

We simulated exponential light pulses with different decay times, $\tau_{d}$, incident in a SiPM with $N_{\mathrm{pix}}=100$. We used the simplified electrical model while assuming $\tau_{\mathrm{rec}}=29$ ns. The trigger probability $P_{\mathrm{trig}}$ was modeled with $U_{\mathrm{ch}}=2.68$ V and $U_{0}=0$ V, as in the simulation case of Figure 4. The operation overvoltage $U_{\mathrm{op}}$ was varied over a wide range from $0.01\;U_{\mathrm{ch}}$ to $100\;U_{\mathrm{ch}}$. Since the response expressed, in terms of the variables *x* and *y*, is independent of both the gain and the absolute value of the PDE, we set G=ε=1 for all $U_{\mathrm{op}}$ values. Correlated noise was ignored for these simulation tests. The results of *y* versus *x* are shown as continuous lines in the upper plot of Figure 5. The lower plot shows the nonlinearity, defined as 1−y/x. For $\tau_{d}/\tau_{\mathrm{rec}}=100$, nonlinearity is smaller than 5% up to x=11. The smaller $\tau_{d}/\tau_{\mathrm{rec}}$, the smaller the *x* value at which a nonlinearity of 5% is reached. We observed that *y* exhibits a logarithmic-like behavior with *x*. In the limit $\tau_{d}\ll \tau_{\mathrm{rec}}$, the results tend toward Equation (Equation 12), where nonlinearity reaches 5% at an *x* value as small as 0.1. The results were checked to not depend on the particular values of $\tau_{d}$ and $\tau_{\mathrm{rec}}$ for a fixed $\tau_{d}/\tau_{\mathrm{rec}}$ ratio.

### 3.3. Operation Overvoltage

As explained in Section 2.2, the trigger probability $P_{\mathrm{trig}}$ in a recovering pixel is assumed to be given by Equation (Equation 9), where *U* recovers to $U_{\mathrm{op}}$ with the time elapsed from the last avalanche as given by Equations (Equation 1) and (Equation 2). In the limit $U_{\mathrm{op}}\gg U_{\mathrm{ch}}$, the trigger probability tends to be a Heaviside step function of time, that is, it recovers to unity almost instantaneously after each avalanche. Therefore, in this limit, nonlinearity is only due to the incomplete recovery of the charge multiplication in subsequent avalanches, given by Equation (Equation 6). In the opposite limit $U_{\mathrm{op}}\ll U_{\mathrm{ch}}$ (assuming $U_{0}=0$ V), both the trigger probability and the charge multiplication are proportional to *U*. In this situation, nonlinearity would be stronger, although operating a SiPM at such low overvoltage is unusual because the PDE is too low.

If $U_{0}=0$ V, the difference in the recovery of the trigger probability and the charge multiplication can be parameterized using the ratio $U_{\mathrm{op}}/U_{\mathrm{ch}}$. This is illustrated by solid lines in Figure 5. The impact of $U_{\mathrm{op}}/U_{\mathrm{ch}}$ on the nonlinearity is not as high as that of $\tau_{d}/\tau_{\mathrm{rec}}$, but it cannot be ignored. Again, the results were checked to not depend on $U_{\mathrm{op}}$ or $U_{\mathrm{ch}}$ for a fixed $U_{\mathrm{op}}/U_{\mathrm{ch}}$ ratio. Remarkably, it can be observed that increasing $U_{\mathrm{op}}/U_{\mathrm{ch}}$ has an effect equivalent to a slight increase in $\tau_{d}/\tau_{\mathrm{rec}}$. This proves that nonlinear effects are essentially due to the balance between the photon rate and the pixel recovery, with the shape of the curve *y* vs. *x* remaining independent of the specific details of the recovery process.

If $U_{0}$ is significant, $P_{\mathrm{trig}}=0$ in the first stage of the pixel recovery while $U\leq U_{0}$, causing the nonlinearity to be stronger. The simulation results for $U_{\mathrm{op}}/U_{\mathrm{ch}}=0.01$ and $U_{0}/U_{\mathrm{op}}=0.5$ are shown as dashed lines in Figure 5 to illustrate this effect.

### 3.4. Correlated Noise

Correlated noise can be understood as a process causing the mean output charge to increase for the same number of incident photons. Therefore, its main effect is that the effective gain of the SiPM in the linear region is greater than the nominal gain *G*, meaning that the response is increased by an extra gain factor that we refer to as 1+c, that is,
(18)$$y_{\mathrm{lin}}=\left(1+c\right)x\;.$$

Notice that correlated noise is stochastic, so this gain factor is associated with an excess noise factor (ENF).

Both crosstalk components have a larger contribution to this gain factor than afterpulsing for the same probability. This is because the mean multiplication charge of avalanches due to afterpulsing is smaller, as they are usually produced while the pixel is still recovering. As an example, we obtained *c* through simulation for a SiPM with $N_{\mathrm{pix}}=100$, $\tau_{\mathrm{rec}}=\tau_{\mathrm{bulk}}=10$ ns, and $U_{0}=0$ and setting $U_{\mathrm{op}}/U_{\mathrm{ch}}=1$. For a crosstalk probability of 10% (irrespective of whether it is prompt or delayed crosstalk) leads to c=0.100, whereas c=0.065 is obtained for the same probability of afterpulsing. In the case of crosstalk, *c* is similar to the mean number of crosstalk seeds $\lambda_{\mathrm{ct}}=0.105$, given by Equation (Equation 16), but they are not identical for several reasons. Firstly, several crosstalk seeds can be generated in the same neighboring pixel, generally resulting in one only avalanche (always for prompt crosstalk). Secondly, fewer crosstalk seeds are generated when the primary pixel is at a border of the SiPM, as it has fewer neighboring pixels. This border effect is less important when the number of pixels is large (e.g., c=0.111 is obtained for $N_{\mathrm{pix}}=3600$ and $P_{\mathrm{ct}}=10\%$). Lastly, cascades of secondary avalanches due to any component of correlated noise contribute to increasing *c*, with this effect being relevant when the probability of correlated noise is high, as discussed in [14].

Analytical expressions of *c* and the associated ENF for prompt crosstalk in the linear region are available in [12,13]. In particular, our MC treatment of prompt crosstalk is equivalent to the four-nearest neighbor model reported in [12], with the difference being that the simulations properly account for both the above-mentioned border effects and cascade processes at any order. However, when delayed crosstalk and afterpulsing are significant, an accurate calculation of *c* should be performed through simulation due to the greater complexity of these processes.

At a high photon rate, where nonlinearity is significant, the incomplete recovery of pixels reduces the gain factor due to correlated noise. Prompt crosstalk was found to be more suppressed than delayed crosstalk and afterpulsing for the same probability. This is because seeds due to prompt crosstalk are produced at the same time as seeds due to photons. In contrast, seeds due to delayed crosstalk and afterpulsing can be produced when the photon rate is lower and the recovery of pixels is more advanced. These effects are more noticeable when studying the shape of the output signal rather than the integrated charge. In Figure 6, averaged simulated signals for exponential pulses with $\tau_{d}=5$ ns and a probability of correlated noise of 20% (assuming a sole noise component in each case) are compared with those obtained in the absence of correlated noise. We used the simplified electrical model with $N_{\mathrm{pix}}=100$, $U_{\mathrm{op}}/U_{\mathrm{ch}}=1$, $U_{0}=0$ V, and $\tau_{\mathrm{rec}}=\tau_{\mathrm{bulk}}=10$ ns. Prompt crosstalk gives rise to an almost constant scale factor to the signal at x=0.5, but the effect is much lower at x=5 as a consequence of nonlinearity. Both afterpulsing and delayed crosstalk produce a lengthening of the output current pulse, which can be significant if $\tau_{\mathrm{bulk}}$ is large. This lengthening, along with the distortion due to nonlinearity, makes the output signal pulse to be flatter than the incident light pulse.

### 3.5. Pulse Shape

We also studied the influence of the shape of the incident light pulse on nonlinearity by comparing the SiPM response for exponential, rectangular, triangular, and double-exponential pulses. For a direct comparison, pulses should have the same mean photon rate $N_{\mathrm{ph}}/\tau$ for a given number $N_{\mathrm{ph}}$ of photons per pulse. The characteristic pulse width τ is
(19)$$\frac{1}{\tau }=\int_{0}^{\infty }p^{2}\left(t\right)\;dt\;,$$
where p(t) is the probability density distribution (PDF) of the photon arrival time. Both the PDF and mean photon rate for the four pulse shapes are shown in Table 2.

Simulation results for different pulses with the same mean photon rate $N_{\mathrm{ph}}/\tau$, setting τ=60 ns, are shown as solid lines in Figure 7. We included exponential pulses of decay time $\tau_{d}=30$ ns, rectangular pulses of width $\tau_{w}=60$ ns, rising triangular pulses with $\tau_{r}=80$ ns and $\tau_{f}=0$, falling triangular pulses with $\tau_{r}=0$ and $\tau_{f}=80$ ns, and double-exponential pulses with $\tau_{1}=10$ ns and $\tau_{2}=20$ ns. The PDF for these pulses is shown in the inset of the figure for a better comparison. For these simulations, we used the simplified electrical model, with $N_{\mathrm{pix}}=100$, $\tau_{\mathrm{rec}}=30$ ns, $U_{\mathrm{op}}/U_{\mathrm{ch}}=1$, $U_{0}=0$ V, and neglected correlated noise. The results are similar for all the pulse shapes, but nonlinearity is stronger for triangular and rectangular pulses, which have a finite duration. Although the mean photon rate is the same for all pulses, nonlinearity also depends on the remaining moments of the distribution. We observed that the smaller the variance of the photon rate (e.g., $\sigma^{2}=0$ for rectangular pulses), the higher the nonlinearity. Interestingly, nonlinearity is slightly stronger for falling triangular pulses (positive skewness) than for rising triangular pulses (negative skewness), as a consequence of the asymmetry introduced by the pixel recovery.

Further results for a larger *x* range from 0 to 1000 and different $\tau /\tau_{\mathrm{rec}}$ ratios are shown as red solid lines in Figure 8. The results for both pure exponential pulses with $\tau_{d}=\tau /2$ and rectangular pulses with $\tau_{w}=\tau$ are shown in the left-hand plot and the right-hand plot, respectively. Again, we used the simplified electrical model, with $N_{\mathrm{pix}}=100$, $U_{\mathrm{op}}/U_{\mathrm{ch}}=1$, $U_{0}=0$ V, and neglected correlated noise. Notice that the SiPM response saturates at a certain level when increasing *x* for rectangular pulses. We checked that the same behavior is shown for triangular pulses. On the other hand, the response is found to have a logarithmic growth with *x* for exponential-like pulses (including double-exponential ones), except in the limit $\tau \ll \tau_{\mathrm{rec}}$. This is due to the fact that, for exponential-like pulses, a few photons may arrive after long periods of times, producing charge avalanches in fairly recovered pixels.

### 3.6. Electrical Model

In the simulation cases discussed in Section 3.2, Section 3.3, Section 3.4 and Section 3.5, we utilized the simplified electrical model described in Section 2.1. Under these simplifications, results expressed in terms of the variables *x* and *y* are independent of the number of pixels $N_{\mathrm{pix}}$, except for the above-mentioned small border effects on crosstalk. However, when $N_{\mathrm{pix}}$ is increased, the sum output current is higher for the same *x* value and gain. As a consequence, the voltage drop across the readout-circuit resistance $R_{s}$ cannot be ignored anymore, and the overvoltage at the pixels is reduced, as discussed in [10,24]. The parasitic capacitance due to the unfired pixels connected in parallel to the fired pixels may also be relevant for large $N_{\mathrm{pix}}$ values, increasing the recovery time, and hence contributing to nonlinearity as well.

Simulation results incorporating the detailed electrical model developed in [24] are depicted as dashed lines in Figure 7. We assumed $N_{\mathrm{pix}}=3600$, a readout-circuit resistance $R_{s}=15\;\Omega$, and the same parameters for this electrical model as in the simulations shown in Figure 3. Nonlinearity becomes significantly more pronounced when the readout-circuit resistance is considered. For instance, the response at x=10 is reduced by approximately 10 to 15 percent for all pulse shapes. However, the resulting curves maintain a shape very similar to those obtained using the simplified electrical model (continuous lines). This suggests that the SiPM response may be effectively described by a universal function depending on a few parameters, as described below. When considering a small number of pixels ($N_{\mathrm{pix}}=100$), the simulation results remained nearly independent of whether the detailed electrical model or the simplified one was employed.

## 4. Analytical Model

In [14], a statistical model including the main factors affecting the nonlinearity (i.e., photon rate, overvoltage, correlated noise, and pulse shape) was developed. From this model, the following simple expression was derived for the responses of SiPMs at moderate nonlinearity:(20)$$y=\frac{1}{1-\gamma }\;\left\{1-\exp\left[-(1-\gamma )(1+c)\;x\right]\right\}\;,$$
where 1+c is the gain factor due to correlated noise explained in Section 3.4, and the dimensionless parameter γ is given by
(21)$$\gamma =2\int_{0}^{\infty }\int_{0}^{\infty }p\left(t\right)\;p\left(t+t_{s}\right)\;P_{\mathrm{trig}}\left(U\left(t_{s}\right)\right)\;\frac{U(t_{s})}{U_{\mathrm{op}}}\;dt_{s}\;dt$$

Here, *t* is the arrival time of a first seed in a pixel, $t_{s}$ is the time of a second seed in the same pixel with respect to the first one, and p(t) stands for the PDF of the output signal pulse, including the lengthening caused through afterpulsing and delayed crosstalk (see Figure 6). The parameter γ is defined such that $\gamma \langle Q_{1}\rangle$ is the mean charge due to a seed generated in a recovering pixel.

The model behaves correctly in the limit situations of very short and long pulses relative to $\tau_{\mathrm{rec}}$, where the SiPM response is known a priori. For very short pulses, γ∼0, and model (Equation 20) reduces to Equation (Equation 12) in the absence of correlated noise. For very long pulses, γ∼1, and model (Equation 20) reduces to Equation (Equation 18). In a general case, model (Equation 20) behaves as Equation (Equation 18) at small *x* values and saturates at 1/(1−γ) at large *x* values. Both *c* and γ can be obtained by fitting model (Equation 20) to experimental data, rather than calculating them from more basic parameters that may not be well known. Indeed, neither the number of seeds per pixel *x* is generally ascertainable in practical situations, because it would require knowing both the number of incident photons and the absolute value of the PDE of the particular SiPM. Therefore, for the application of the model, the product of (1+c) and the conversion factor from the light pulse intensity to *x* can be used as a fitting parameter.

In Figure 8, fits of the model (Equation 20) to simulated data for pure exponential pulses and rectangular pulses are shown as blue dotted lines in the left-hand plot and right-hand plot, respectively. We fixed c=0, because correlated noise was ignored in the simulations. Although results are shown up to x=1000, these fits were performed in a limited range of *x*, depending on $\tau /\tau_{\mathrm{rec}}$, where nonlinearity is weak. Notice that γ accounts for interactions between only two seeds in the same pixel; hence, this model is expected to be valid only for x≲2. Indeed, the exponential saturation predicted by the model is actually not observed in the simulated data, especially for exponential-like pulses.

We extended the model (Equation 20) to arbitrary *x* values based on our analysis of simulation results. In addition to addressing the limit situations for very short and very long pulses, the new model should be capable of reproducing either the logarithmic behavior for exponential-like pulses or the smooth saturation observed in simulated data for finite pulses. The model should also include the suppression of correlated noise with increasing *x*, as described in Section 3.4. Given these conditions, we propose the following ansatz for exponential-like pulses:(22)$$y=\left[1+c\;e^{-dx}+a_{e}\;\ln\left(1+b_{e}\;x\right)\right]\left(1-e^{-x}\right)\;,$$
and the following for finite pulses:(23)$$y=\left[1+c\;e^{-dx}+\frac{a_{f}\;x}{1+b_{f}\;x}\right]\left(1-e^{-x}\right)\;,$$
where we used the indexes “e” and “f” to distinguish between the two categories of pulses.

The meanings of the different terms in the above expressions are as follows:The factor $1-e^{-x}$ in both models is identified with the mean fraction of fired pixels per pulse in the absence of crosstalk, Equation (Equation 12).The term $1+c\;e^{-dx}$ within the brackets is the mean contribution of the first photon detected in a fired pixel, including the correlated noise that it induces in this pixel and the neighboring ones. This term tends to 1+c in the linear region (i.e., x→0), where *c* has the same meaning as in model (Equation 20). The parameter *d* accounts for the suppression rate of correlated noise with increasing *x*.A term, either $a_{e}\ln\left(1+b_{e}\;x\right)$ or $\frac{a_{f}\;x}{1+b_{f}\;x}$, accounts for all the remaining avalanches produced during the pixel recovery. This term is small when the pulse width is short relative to $\tau_{\mathrm{rec}}$.

We can interpret the meaning of the pair of parameters $\left(a_{e},\;b_{e}\right)$ or $\left(a_{f},\;b_{f}\right)$ by separately analyzing the regimes of low and high nonlinearity. On the one hand, the models (Equation 20), (Equation 22), and (Equation 23) are mutually equivalent when considering small *x* values. In this regime, the relationships among the models, given by comparing the first two terms of their Taylor series at x=0 in the absence of correlated noise, can be expressed as follows:(24)$$a_{e}b_{e}\approx a_{f}\approx \frac{\gamma }{2}\;.$$

On the other hand, the asymptotic behavior of models (Equation 22) and (Equation 23) as x→∞ is quite different, where
(25)$$\begin{array}{cc} \lim_{x\to \infty }a_{e}\ln\left(1+b_{e}\;x\right) & =a_{e}\ln\left(x\right)\;, \end{array}$$
(26)$$\begin{array}{cc} \lim_{x\to \infty }\frac{a_{f}\;x}{1+b_{f}\;x} & =\frac{a_{f}}{b_{f}}\;. \end{array}$$

Fits of the models (Equation 22) and (Equation 23) to simulated data are shown as black dashed lines in Figure 8. We set c=d=0, because correlated noise was ignored in the simulations. Both models fit adequately to simulated data in the entire *x* range from 0 to 1000, with residuals being generally less than 5%, although it depended on both the ratio of the pulse width to $\tau_{\mathrm{rec}}$ and the selected fitting range. For instance, the maximum residual for exponential pulses with $\tau_{d}/\tau_{\mathrm{rec}}=1$ is 10% when fitting Equation (Equation 22) in the entire range, but the maximum residual is as low as 3.4% when the fit is limited to *x* values smaller than 10, where nonlinearity reaches 70% (see Figure 5). In this reduced *x* range, both models (Equation 22) and (Equation 23) provide very similar results. The fit is poorer for $\tau_{d}/\tau_{\mathrm{rec}}=100$, where nonlinearity is only noticeable at very large *x* values. Similar residuals are obtained for double exponential pulses as well as when fitting the model (Equation 23) for triangular pulses.

Fits to simulation results incorporating correlated noise are shown in the left-hand plot of Figure 9. For these simulations, we used short exponential pulses with $\tau_{d}/\tau_{\mathrm{rec}}=0.1$ and assumed the simplified electrical model with $U_{\mathrm{op}}/U_{\mathrm{ch}}=1$, $U_{0}=0$ V, and a probability of 20% of prompt crosstalk, delayed crosstalk, or afterpulsing. We plotted the ratio $y/(1-e^{-x})$, which represents the mean contribution per fired pixel in order to visualize the contributions from the two terms within the brackets in model (Equation 22). The isolated contribution from the first term (i.e., the contribution from the first detected photon and the correlated noise that it induces) is also depicted as a point–dash line in each case. When fitting model (Equation 22) for *x* values smaller than 10, the residuals are below 1%. Notice that, although *c* is larger for crosstalk than for afterpulsing for the same probability, crosstalk is suppressed more rapidly as *x* increases. This results in a smaller response in the case of crosstalk compared to afterpulsing at large *x* values. If crosstalk is low, the impact of the suppression of the correlated noise on the SiPM response may be ignored, and the parameter *d* can be set to 0.

Models (Equation 22) and (Equation 23) fit well to both simulated and experimental data for all the variety of cases studied in this work (e.g., different $U_{\mathrm{op}}/U_{\mathrm{ch}}$ values and probabilities of correlated noise). As an example, fits of model (Equation 22) to experimental data from [14] are depicted as dotted lines in Figure 4. For these fits, we assumed the same conversion factors fε from the number of photons to the number of seeds as those used in the simulations. While it is challenging to find input parameters for the simulation that precisely match the experimental data, the fit of the model (Equation 22) is very good, with deviations within experimental errors for all curves.

We also fitted model (Equation 23) to experimental data from [32] for three different SiPMs: PM3350 from Ketek GmbH [33] (pixel size of 50μm, $N_{\mathrm{pix}}=3600$, and $\tau_{\mathrm{rec}}\approx 83$ ns), Hamamatsu S10362-33-050C (pixel size of 50μm, $N_{\mathrm{pix}}=3600$, and $\tau_{\mathrm{rec}}\approx 20$ ns), and Hamamatsu S10362-33-100C (pixel size of 100μm, $N_{\mathrm{pix}}=900$, and $\tau_{\mathrm{rec}}\approx 48$ ns). The SiPMs were illuminated by rectangular-shaped blue LED pulses with different widths from 12 to 73 ns. The SiPM response was characterized in terms of the ratio $\frac{\langle Q\rangle }{\langle Q_{1}\rangle }$ as a function of the mean number of incident photons per pulse. In Figure 10, we reproduce the experimental data for pulse widths of 35 and 73 ns as curves *y* versus *x*, where the variable *x* was obtained from Equation (11), knowing the PDE. For the Hamamatsu S10362-33-100C SiPM, we utilized the PDE value ε=0.35 at $U_{\mathrm{op}}=1.3$ V, as reported in [32]. However, the authors did not provide the PDE values for the other two SiPMs. In these cases, we determined ε(1+c) from the experimental data in the linear region, Equation (Equation 18). Subsequently, we estimated ε and *c* separately to be consistent with the data available in the literature. In particular, for the Hamamatsu S10362-33-050C SiPM, we obtained ε(1+c)=0.28 and estimated ε=0.17 and c=0.65 at $U_{\mathrm{op}}=1.65$ V, which align with the data reported in [30]. Finally, the model (Equation 23) was fitted to the experimental curve *y* versus *x*, fixing the parameter *c*. The fit residuals were generally smaller than 5% for all the curves, and they remained low for other reasonable assumptions of ε and *c*. For instance, in assuming ε=0.21 and c=0.33 for the Hamamatsu S10362-33-050C SiPM, the residuals were still smaller than 10%. Therefore, our model can provide an adequate description of the SiPM response as a function of the light pulse intensity even when an accurate PDE value at the given wavelength is not available.

We observed that the effect of the suppression of correlated noise is noticeable in these experimental data from [32]. To illustrate this, the curves $y/(1-e^{-x})$ versus *x* for the Hamamatsu S10362-33-050C SiPM for both pulse widths are shown in the right-hand plot of Figure 9. The fits of model (Equation 23) are depicted as dashed lines, showing a good agreement (residuals smaller than 4%). Notice that the larger the pulse width, the more important the term $\frac{a_{f}\;x}{1+b_{f}\;x}$. Therefore, the parameter *d* can be set to 0 to fit the SiPM response for long light pulses.

## 5. Conclusions

In this work, we carried out a systematic study on the nonlinear responses of SiPMs for light pulses, identifying and parameterizing the main factors affecting nonlinearity. For this purpose, we used the MC code developed in [24], with some improvements in the description of the correlated noise and the pixel recovery. Simulations were shown to reproduce experimental data on the output charge for scintillation light pulses as a function of both the pulse intensity and the SiPM operation overvoltage.

We characterized the SiPM response in terms of the variables *y* and *x*, defined, respectively, as the effective number of full-charge avalanches per pixel, Equation (Equation 10), and the expected number of avalanches per pixel for an ideal SiPM in the absence of nonlinearity and correlated noise, Equation (11). We found that the shape of the curve *y* versus *x* is quite universal.

Nonlinearity essentially depends on the balance between the photon rate and the pixel recovery time $\tau_{\mathrm{rec}}$ after a charge avalanche, which can be parameterized as the ratio of the pulse width τ (e.g., the decay time for an exponential pulse) to $\tau_{\mathrm{rec}}$ for a given *x* value. For $\tau /\tau_{\mathrm{rec}}\approx 1$, nonlinearity reaches around 5% at *x* values as low as 0.2.

When a SiPM is operated at a high overvoltage $U_{\mathrm{op}}$, such that the PDE is close to its maximum value, the trigger probability of a pixel recovers faster than the pixel overvoltage, reducing nonlinearity in a similar way as if $\tau_{\mathrm{rec}}$ was made shorter. We parameterized it as the ratio of the operation overvoltage to the characteristic parameter $U_{\mathrm{ch}}$ of the PDE vs. $U_{\mathrm{op}}$ curve assuming Equation (Equation 7), albeit other parameters should be used instead in case the overvoltage dependence of the PDE is described using a different function (e.g., for red- and NIR-sensitive SiPMs).

We also studied the SiPM response for different light pulse shapes, finding that nonlinearity is stronger for finite pulses (e.g., rectangular and triangular ones) than exponential-like pulses with the same mean photon rate. The level of nonlinearity depends on the particular pulse shape. In addition, we demonstrated that the SiPM response saturates when increasing the light pulse intensity for finite pulses, whereas it shows a logarithmic growth for exponential-like ones.

The correlated noise increases the effective gain of the SiPM in the linear region, with a minor influence on nonlinearity. In particular, prompt crosstalk was found to be suppressed rapidly at increasing photon rates. However, afterpulsing and delayed crosstalk may still be relevant for intense light pulses, because these noise components lead to a lengthening of the output signal. We showed how the signal is distorted through nonlinearity and correlated noise. The impact of the details of the electrical model on the simulation results was also studied, and the results are in agreement with the findings shown in [24].

Based on these simulation results and the statistical model developed in [14], we found the two simple expressions (Equation 22) and (Equation 23) that correctly reproduce the features of the SiPM response for exponential-like and finite light pulses, respectively. Even though these models have only four fitting parameters, they were shown to be valid even for light pulses of very high intensity and for a wide range of situations. In the cases where the impact of the suppression of correlated noise at an increasing photon rate is low (e.g., low crosstalk and long light pulses), the number of fitting parameters can be reduced to three. The models provide insights into the different contributions to nonlinearity.

## Figures and Tables

**Figure 1 sensors-24-02648-f001:**
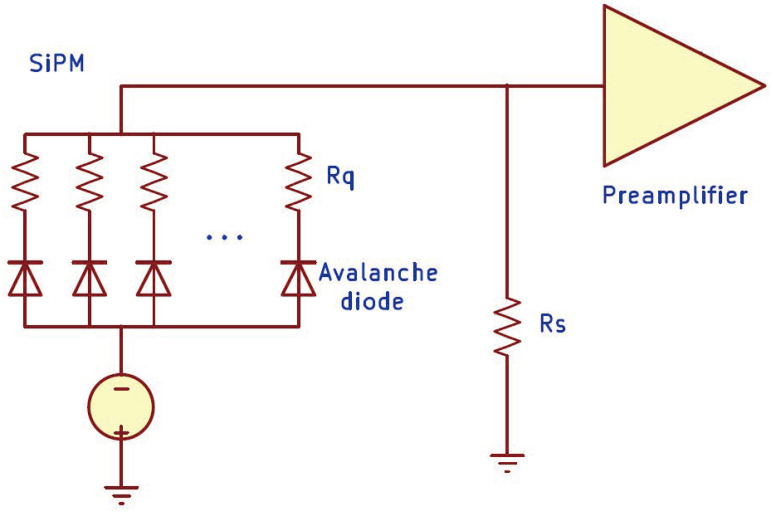
Schematic of a SiPM and its readout circuit.

**Figure 2 sensors-24-02648-f002:**
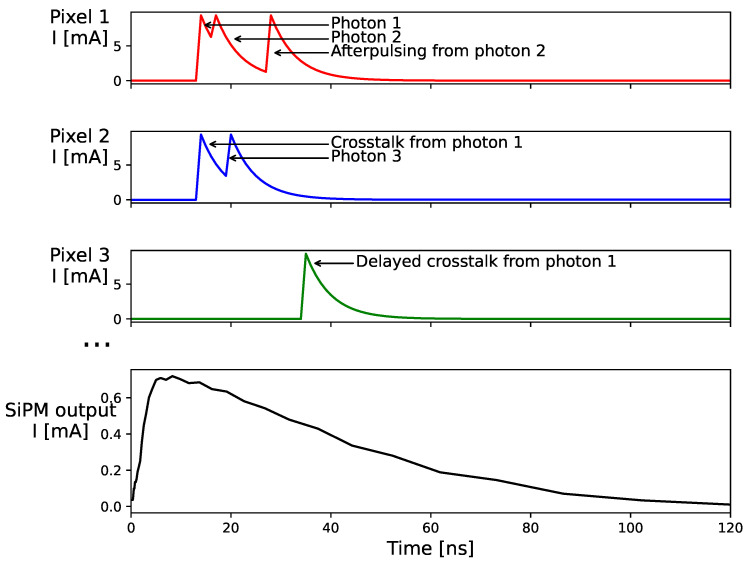
Simulated current signals generated by three pixels of a SiPM and total output signal. The origin of each avalanche (i.e., photon, crosstalk, or afterpulsing) is tagged in the simulation, although avalanches are indistinguishable from each other. The effect of the pixel recovery in the avalanche amplitude can be appreciated in the signals from pixels 1 and 2.

**Figure 3 sensors-24-02648-f003:**
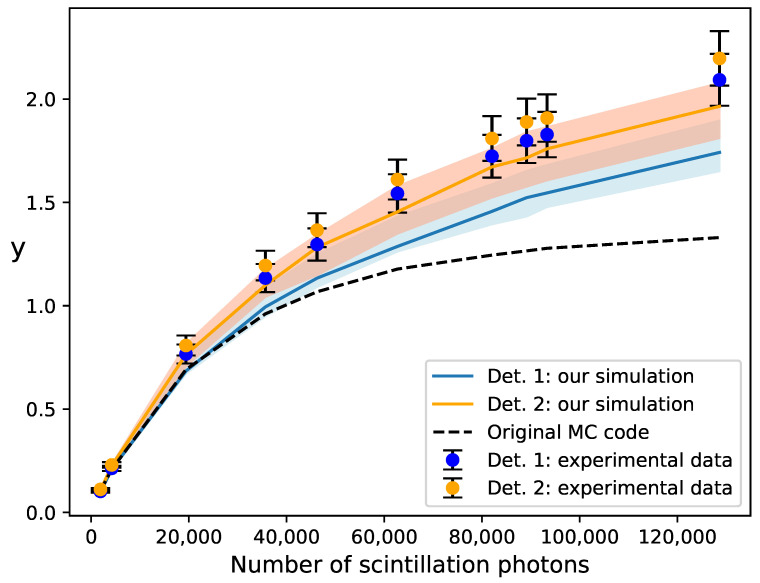
Comparison of simulation results with experimental data for two Hamamatsu S10362-33-050C SiPMs coupled to LaBr3:5%Ce scintillators from [18]. Simulation results obtained with the original MC code from [24] are represented by the dashed line. The results obtained with our modified version of the code are represented by the continuous lines, where the color bands illustrate the variation in our simulations when the input parameters are varied within their uncertainties. The SiPM response is expressed in terms of the variable *y*, defined by Equation (Equation 10), while the response was expressed in terms of the variable $\chi =\frac{\langle Q\rangle }{\langle Q_{1}\rangle N_{\mathrm{seed}}}=\frac{y}{x}$ in Figure 7 of [24].

**Figure 4 sensors-24-02648-f004:**
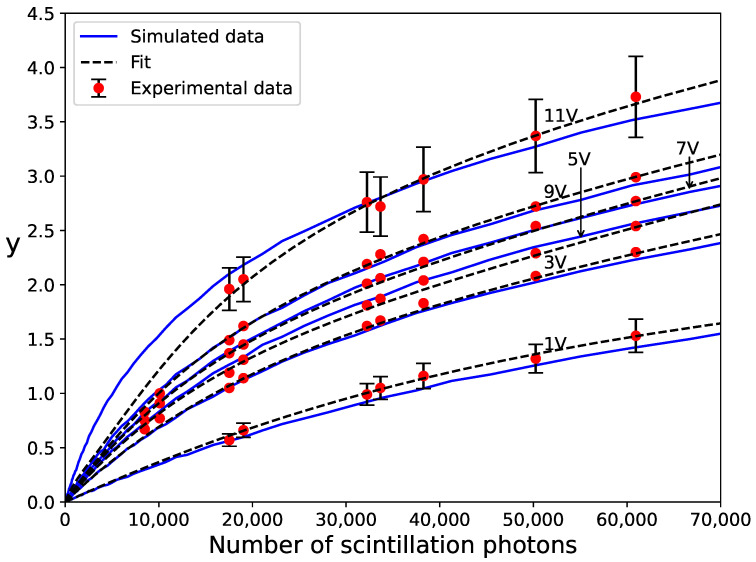
Comparison of simulation results with experimental data for a Hamamatsu S13360-1350CS SiPM coupled to a LYSO(Ce) scintillator, taken from [14]. Fits of model (Equation 22) to experimental data are also shown as dashed lines.

**Figure 5 sensors-24-02648-f005:**
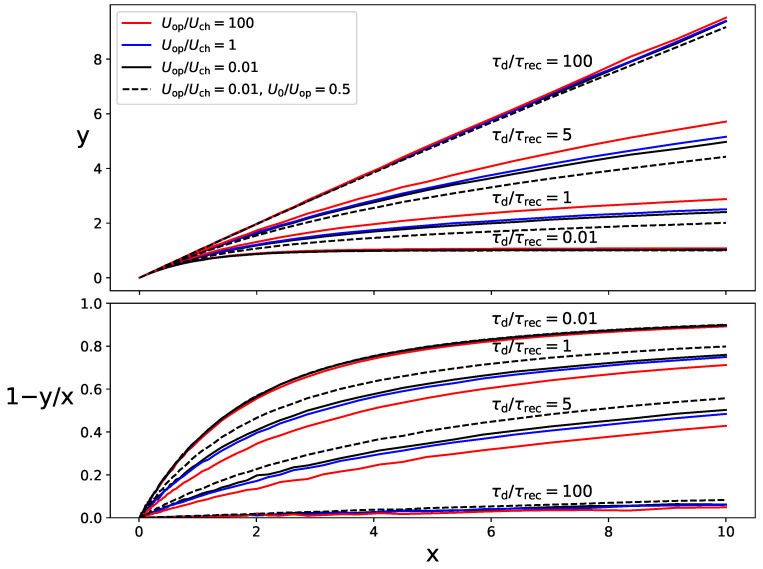
Simulation results for different values of $\tau_{d}/\tau_{\mathrm{rec}}$, $U_{\mathrm{op}}/U_{\mathrm{ch}}$, and $U_{0}/U_{\mathrm{op}}$ for exponential light pulses. The SiPM response (upper plot) is expressed in terms of the variables *y* and *x* defined by Equations (Equation 10) and (11), respectively. In the lower plot, the nonlinearity parameter 1−y/x is shown.

**Figure 6 sensors-24-02648-f006:**
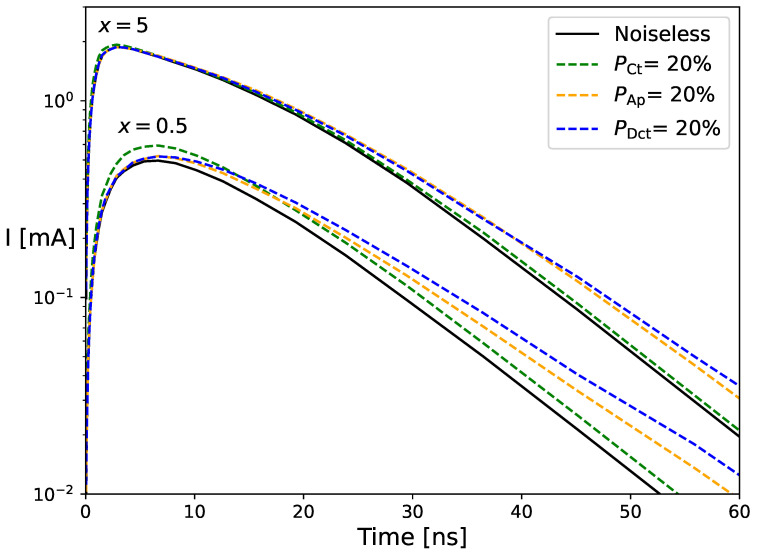
Averaged SiPM signals simulated for different situations of correlated noise (i.e., no noise, 20% of prompt crosstalk, 20% of afterpulsing, or 20% of delayed crosstalk) and different numbers of seeds per pixel (i.e., x=0.5 and 5).

**Figure 7 sensors-24-02648-f007:**
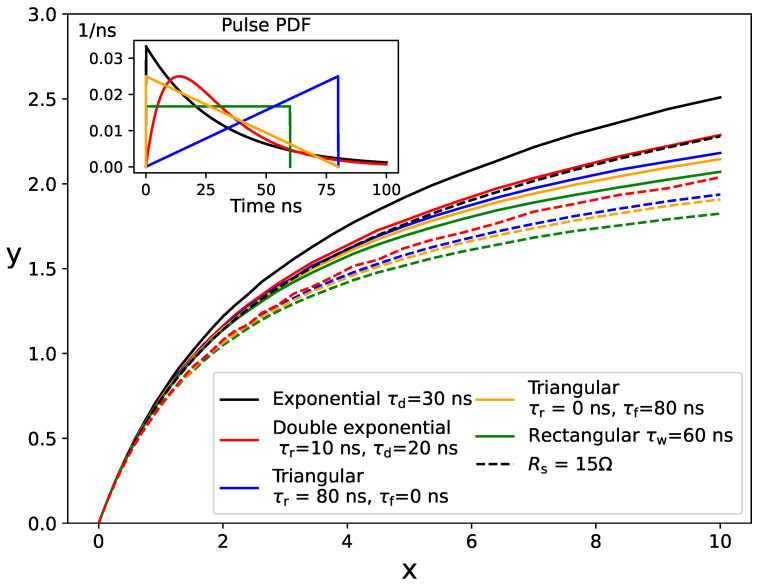
Simulated SiPM responses for light pulses of different shapes but the same mean photon rate. The inset plot shows the probability density functions of the light pulses used in the simulations. The continuous lines represent the results utilizing the simplified electrical model described in Section 2.1. The dashed lines represent the results incorporating the detailed electrical model described in [24], assuming an input impedance of the readout electronics of $R_{s}=15\;\Omega$.

**Figure 8 sensors-24-02648-f008:**
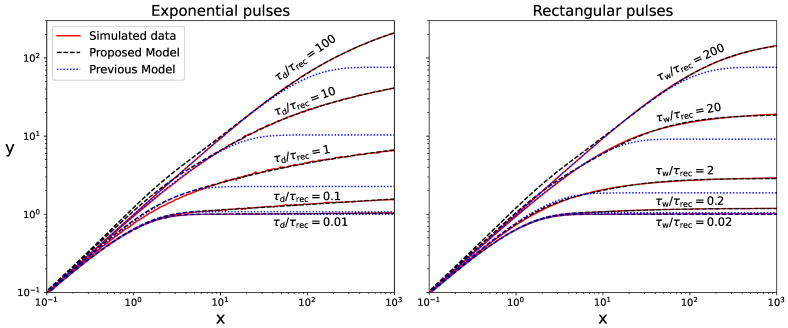
Simulation data for both exponential pulses with different $\tau_{d}/\tau_{\mathrm{rec}}$ ratios (left-hand plot) and rectangular pulses with different $\tau_{w}/\tau_{\mathrm{rec}}$ ratios (right-hand plot). Correlated noise was ignored. Fits of the model (Equation 20) from [14] and either model (Equation 22) or (Equation 23) proposed in this work for the two types of pulses are also shown. The results are depicted for a large range of *x*, but the model (Equation 20) was fitted in a limited range where nonlinearity is still moderate. The models (Equation 22) and (Equation 23) outperform the model (Equation 20) for both types of pulses.

**Figure 9 sensors-24-02648-f009:**
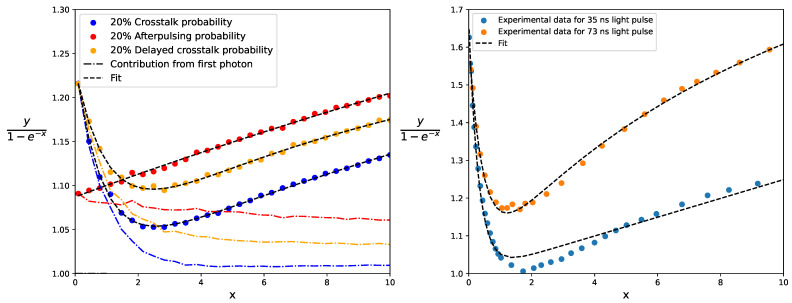
(**Left**): Simulation results for short exponential pulses for a SiPM with 20% of prompt crosstalk, 20% of delayed crosstalk, or 20% of afterpulsing. The point–dash lines represent the isolated contribution from the first photon detected by a pixel, including the correlated noise that it induces. Continuous lines represent the best fits of model (Equation 22). (**Right**): Experimental data from [32] for a Hamamatsu S10362-33-050C illuminated by LED pulses of different widths. Fits to model (Equation 23) are shown.

**Figure 10 sensors-24-02648-f010:**
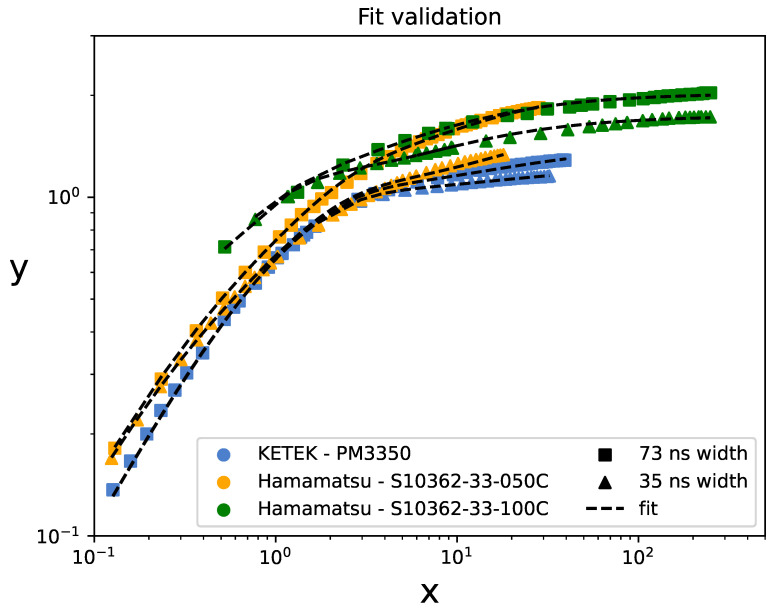
Fits of model (Equation 23) to experimental data from [32] for three different SiPMs illuminated by LED pulses of different widths.

**Table 1 sensors-24-02648-t001:** Input parameters used in [24] to simulate the experiment described in [18].

Detector	$U_{\mathrm{op}}$ (V)	G (10^5^)	$R_{q}$ (kΩ)	$\lambda_{\mathrm{ct}}$	$\lambda_{\mathrm{aft}}$	$\tau_{\mathrm{aft}}$	fε
Detector 1	1.26±0.05	8.3±0.5	145.3±0.5	0.140±0.05	0.124±0.05	25±8	0.157
Detector 2	1.39±0.05	7.9±0.5	140.4±0.5	0.132±0.05	0.132±0.05	26±8	0.172

**Table 2 sensors-24-02648-t002:** Characteristics of different pulse shapes.

Shape	Probability Density Function	Mean Photon Rate
Exponential	$\frac{1}{\tau_{d}}e^{-t/\tau_{d}}\;\;\left(t\geq 0\right)$	$\frac{N_{\mathrm{ph}}}{2\tau_{d}}$
Rectangular	$\frac{1}{\tau_{w}}\;\;\left(0\leq t\leq \tau_{w}\right)$	$\frac{N_{\mathrm{ph}}}{\tau_{w}}$
Double exp.	$\frac{1}{\tau_{2}-\tau_{1}}\left(e^{-t/\tau_{2}}-e^{-t/\tau_{1}}\right)\;\;\left(t\geq 0\right)$	$\frac{N_{\mathrm{ph}}}{2(\tau_{1}+\tau_{2})}$
Triangular	$\;\;\;\frac{2}{\tau_{r}+\tau_{f}}\frac{t}{\tau_{r}}\left(0\leq t\leq \tau_{r}\right)\;\frac{2}{\tau_{r}+\tau_{f}}\left(1-\frac{t}{\tau_{f}}\right)\left(\tau_{r}<t\leq \tau_{r}+\tau_{f}\right)$	$\frac{4N_{\mathrm{ph}}}{3(\tau_{r}+\tau_{f})}$

## Data Availability

The MC code is available upon request.

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
