# Peer review of "Understanding the Nonlinear Response of SiPMs"

_sensors, 2024, doi:10.3390/s24082648_

Round 1

Reviewer 1 Report

Comments and Suggestions for Authors

The paper “Understanding the Nonlinear Response of SiPMs” presents an extended evaluation of the original author’s analytical model of the nonlinear SiPM response (ref. [14] of the paper). The model allows to account an arbitrary light pulse shapes as well as a variety of correlated noise contributions and describes a mean output charge of the SiPM. The paper is worth publishing as an attempt to improve our understanding of the SiPM nonlinearity in demand for high dynamic range applications.

Regretfully, despite the good quality of the paper in general, it has several drawbacks to be addressed.

1.      Parameterization

a.      The authors define the probability of triggering a charge avalanche by eq (9) Ptrig(U) =ε(U)/ε(Uop).

But eq. (9) is incorrect because it means that Ptrig(U) → 1 for any Uop at the end of the recovery process when U(t) →Uop.

This mistake is especially critical if Uop → Uo and PDE ε(Uop)→ 0 but Ptrig(U →Uop) → 1 anyway.

In fact, Ptrig(U) is typically approximated by eq. (7) or similar as a sole voltage-dependent factor for PDE.

This issue affects all simulations related to correlated effects and pixel recovering processes (e.g. eqs. (16), (19) with Ptrig(U)). 

b.      The authors define the mean number of correlated seeds during recovery as a linear function of U(t) by λ′i = λi U/Up (page 6 line 189) “because the number of emitted infrared photons is proportional to the avalanche multiplication (6).”

But I guess the number of correlated events should also depend on the probability of triggering in a similar manner as the number of photon detection seeds Nseed in eq. (8) is defined to be proportional to PDE and, therefore, to Ptrig. The photon seeds coincide with the detected photons in a linear regime, but what about correlated events?

Authors should clarify definitions of λ and λ′ and differentiation between photon seeds and correlated seeds.

2.      Assumptions

a.      The number of correlated events of any kind is assumed to be of a Poisson distribution (page 6 line 187). The model (ref. [14]) has no statements about their distributions and operates with a parameter c as “the mean value of the relative contribution from secondary avalanches”. In earlier papers [11, 12] the authors evaluated some known distribution models (Geometric, Borel [13]) and their nearest neighbors model to be in good agreement with experimental results. These models are widely utilized in many studies. However, to the best of my knowledge, the Poisson distribution has not been evidenced in publications even if it seems to be a rather natural option.

Authors are advised to present their information on this topic and discuss the influence of the event probability distribution on the model and simulation results.

b.      The authors expressed the delay time distribution for afterpulsing and delayed crosstalk by (13) based on their study [11]. However, most of the papers on this topic evaluated and modeled the correlated event time distributions as a sum of exponential decays with a few different time constants related to different transient processes, say, detrapping from deep level traps of different activation energies in case of afterpulsing. Why did the detrapping times disappear in the model?

Authors are advised to present their information on this topic and discuss the influence of the time probability distribution on the model and simulation results.

Minor remarks

1.        Table 2 “Characteristics of different pulse shapes” is worth clarifying.

It combines PDFs of a random variable - an arrival time of the single non-random photon in the pulse of specific shape. The common mean characteristic of this random variable is the mean arrival time. Some “intensity” is often defined as an inverse of that time x Nph, but it differs twice from “Mean photon rate” expressions. Say, the exponential shape has the mean arrival time equal to τd, the rectangular – τw/2, i.e. not equal to corresponding denominators in Table 2.

2.        Mistyping in Ref. [2] -                       “Lapington, J.S.; Collaboration, C.S….”

Reviewer 2 Report

Comments and Suggestions for Authors

The manuscript reports on a computational study of SiPM nonlinearity. The simulations considered several aspects of the light pulse (intensity, timing and pulse shape) and the main functional parameters of the photodetector.

The simulation is based on a Monte Carlo simulation platform implemented with MATLAB already available in literature, published in 2013 by different authors (reference 24).

In this contribution, the authors expand the original computational method, by considering additional light pulse and photodetector parameters in order to better reproduce the nonlinear effects of SiPM response. Moreover, the authors here propose an empirical fitting formula for SiPM nonlinearity.

The topic of this manuscript is well known and studied in literature, from both an analytical and computational point of view. Nevertheless it is worth studying in depth this topic, in order to provide effective models that may help researchers in the design of complex experiments and applications based on SiPMs.

While the phenomenology of SiPM behavior, in particular nonlinearity, is well described and understood, this work can be in principle of interest for researchers in the field, if it provides new effective tools.

In my opinion, the manuscript in the present form does not provide effective results that can be of practical use and requires some modifications and clarifications.

1. The improvements obtained in this work compared to the original Monte Carlo method of reference 24 are not stated clearly. Figure 3 reports a comparison with the original method, but the same plot was not reported in the original paper, thus it should be explained better how the results were extrapolated from the original work. In addition, the authors should clarify which ones among the different modifications to the original code provided the most significant difference with the original results. In order to make a more clear comparison with the original method, it is important to show clearly the effectiveness of each improvement to the code.

2. Equations 20 and 21 provide a good fitting to interpolate simulation results, but cannot be used without a significant set of simulation data. In order to obtain a fitting formula of practical use, it would be necessary to understand how SiPM functional parameters play a role in determining the fitting parameters a and b. This result would be a real advance in SiPM nonlinearity understanding, while this work in its present form requires scholars to run the simulation code themselves with the hope that it applies correctly to their application.

3. In the present form of the manuscript, it is not clear how general the simulation tool can be considered. In the last few years, the main SiPM vendors (manly Hamamatsu, Broadcom and OnSemi) have released on the market new SiPM families, with deep design and technological differences: cell size (50 um to 12 um), cross-talk inhibition technologies, junction polarity and spectral sensitivity. Different technological platforms are used typically in different application fields (LiDAR, PET, cryogenic detectors, ...). Alongside the successful comparison with a single SiPM technology, it would be crucial to understand and discuss the limits of application of the proposed computational method.

Round 2

Reviewer 2 Report

Comments and Suggestions for Authors

The authors have addressed all the points raised at the first stage of the review process. In my opinion, the paper can be accepted for publication in the present form.